# Photocatalytically Active Semiconductor Cu₃P Unites with Flocculent TiN for Efficient Removal of Sulfamethoxazole

Huancong Shi [1,2] , Xulei Yao [1], Shijian Lu [3,*], Yuanhui Zuo [1,2,4,*], Tao Zheng [1] and Liangquan Jia [5,*]

1 School of Energy and Power Engineering, Department of Environmental Science and Engineering, University of Shanghai for Science and Technology, Shanghai 200093, China
2 Research Institute of Fudan University in Ningbo, Ningbo 315204, China
3 Carbon Neutrality Institute, China University of Mining and Technology, Xuzhou 221008, China
4 Huzhou Institute of Zhejiang University, Huzhou 313000, China
5 School of Information Engineering, Huzhou University, Huzhou 313000, China
* Correspondence: lushijian@cumt.edu.cn (S.L.); zyh06101@126.com (Y.Z.); 02426@zjhu.edu.cn (L.J.)

**Abstract:** Sulfamethoxazole is a widely—used antibiotic with high water solubility and low biodegradability, which was considered a refractory environmental pollutant. Hence, a series of functionalized hybrids uniting Cu₃P with TiN were prepared. The Cu₃P/TiN—x composites remarkably removed the sulfamethoxazole in solution compared with Cu₃P and TiN alone. All the as—prepared Cu₃P/TiN—x hybrids integrated the advantages of strong adsorption and photocatalysis and achieved removal rates above 70% of sulfamethoxazole. Among the composites, the Cu₃P/TiN—2 with a 1:1 molar ratio of Cu: Ti reached a 90% removal rate under dark adsorption for 30 min and subsequent photodegradation for 120 min. The enhanced performance of the Cu₃P/TiN—x composites is attributed to the introduced flocculent TiN with a large specific surface area and high conductivity that provide more reactive sites and high electron transferability. Meanwhile, the strong corrosion resistance and chemical stability were also beneficial to the improved performance. Cycling experiments further demonstrate the stability and reliability of the composites. In addition, the capture experiments indicated that the superoxide radical ($\cdot O_2^-$) and hydroxyl radical ($\cdot OH$) played a major role in sulfamethoxazole degradation.

**Keywords:** Cu₃P/TiN; plasmonic photocatalyst; sulfamethoxazole; photodegradation



## 1. Introduction

Antibiotics have been widely used in disease control, agricultural production, animal husbandry, and aquaculture [1]. It is commonly believed that antibiotics have facilitated people's lives and contributed greatly to improving human and animal health [2,3]. However, antibiotic abuse has posed a serious threat to human health and ecosystems since its accumulation in the environment may give rise to drug—resistant pathogens [4,5]. Moreover, some antibiotics may not be absorbed and metabolized after being injected into humans and animals due to excessive consumption. They are then discharged into the surrounding environment directly, causing severe pollution [6,7]. Sulfamethoxazole antibiotic (i.e., SMX) was discovered in the mid—20th century [8–10]. Because of its good antibacterial effect and low price, it has become a drug widely used in the treatment and prevention of various bacterial infections [11–13]. However, the overuse of sulfamethoxazole antibiotics makes it difficult to be absorbed totally by animals and plants, and then accumulates in the environment, leading to obvious immunotoxicity, mutagenicity, embryonic toxicity, and other major toxicological effects threatening public health and the social environment [14,15]. Therefore, the effective removal of SMX in sewage is very significant. Compared with oxidative degradation [16,17], microbial treatment, and phytoremediation, photocatalytic degradation is used in the treatment of pollutants more because of its advantages of high efficiency, lower energy consumption, etc. [18,19]. The development of a

photocatalyst responsive to visible light is an urgent requirement for the efficient degradation of SMX [20,21]. Transition metal phosphides, especially p—type semiconductor $Cu_3P$ with a narrow band gap [22,23], have attracted the attention of researchers due to their potential application in photocatalysis [24]. However, the photo—induced electron—hole pair separation efficiency is still an important limiting factor for its wide application. Recently, the introduction of nanostructure possessing a localized surface plasmon resonance (LSPR) demonstrated a promising approach to largely ameliorate electron—hole recombination. Additionally, titanium nitride (TiN), having gold—like optical properties, possesses a plasmonic resonance absorption peak which can expand the responsive illumination range besides its intrinsic characteristics, including a high melting point and good chemical stability. TiN is a nonmetal conductive ceramic; it exhibits excellent metallic characteristics due to its metallic band structure and high carrier concentrations [25–27]. Moreover, TiN has a regular stable pore structure and a relatively large specific surface area, promoting the transport and absorption of reactant and light, thus contributing to good photocatalytic activity [28]. Therefore, the combination of TiN and $Cu_3P$ is a convincing strategy.

In this work, $Cu_3P$/TiN—x were composite materials. $Cu_3P$/TiN—x were specifically marked as $Cu_3P$/TiN—1, $Cu_3P$/TiN—2, $Cu_3P$/TiN—3, and $Cu_3P$/TiN—4 corresponding to the different Cu/Ti molar ratios, i.e., 1/0.5, 1/1, 1/5, and 1/10, respectively. $Cu_3P$/TiN—x composites were successfully prepared by simple calcination. Some universal characterization methods were used to identify the structure and composition. The newly—made $Cu_3P$/TiN—x composites possess high photocatalytic degradation efficiency and good stability. In order to test the catalytic activity of the synthesized catalysts, SMX was selected as the target refractory pollutant, and the physical characteristics of the functional catalysts were studied in detail, including absorption intensity of visible light, the efficiency of degradation of pollutants and stability of $Cu_3P$/TiN—x. In order to further understand the charge separation efficiency of related materials, a large amount of literature was reviewed. Literature exhibited that both TiN and $Cu_3P$ composites have higher photocurrent density than pure materials. TiN is a metallic interstitial compound, where the nitrogen atoms go to the interstices of the metal—based framework, and thus TiN can exhibit excellent metallic characteristics. $Cu_3P$ semiconductor easily generates electrons under visible light, and the photogenerated electrons can easily transfer to TiN. According to the previous report, TiN, as an effective electron trap, can promote the separation of photogenerated carriers on semiconductors [24,29].

## 2. Results

### 2.1. Characterization Analysis

#### 2.1.1. XRD Patterns

X-ray diffraction (XRD) was used to investigate the crystal structure and phase composition of bare TiN, bare $Cu_3P$, and the $Cu_3P$/TiN—x composites. As seen in Figure 1, the XRD pattern of original TiN exhibits different diffraction peaks at 36.8°, 42.7°, and 62.1°, consistent with (111), (200), and (220) planes, respectively, of TiN (JCPDS No. 87—0632). For $Cu_3P$, the diffraction peaks at 39.1°, 45.1°, and 46.2° correspond to the (202), (300), and (113) planes (JCPDS No. 71—2261). It is worth noting that all the characteristic peaks of TiN and $Cu_3P$ can be identified in $Cu_3P$/TiN—x composites distinctly. In $Cu_3P$/TiN—1 and $Cu_3P$/TiN—2, the peaks according to $Cu_3P$ are more obvious. Correspondingly, the peaks matching with TiN are more dominant in $Cu_3P$/TiN—3 and $Cu_3P$/TiN—4. The foregoing situation is consistent with the Cu/Ti molar ratio setting at the beginning of the material design. $Cu_3P$ composition is dominant at the beginning and gradually changes to TiN composition dominance with the increase of the Ti ratio. These results indicated that $Cu_3P$/TiN—x composites were successfully synthesized in predetermined proportions.

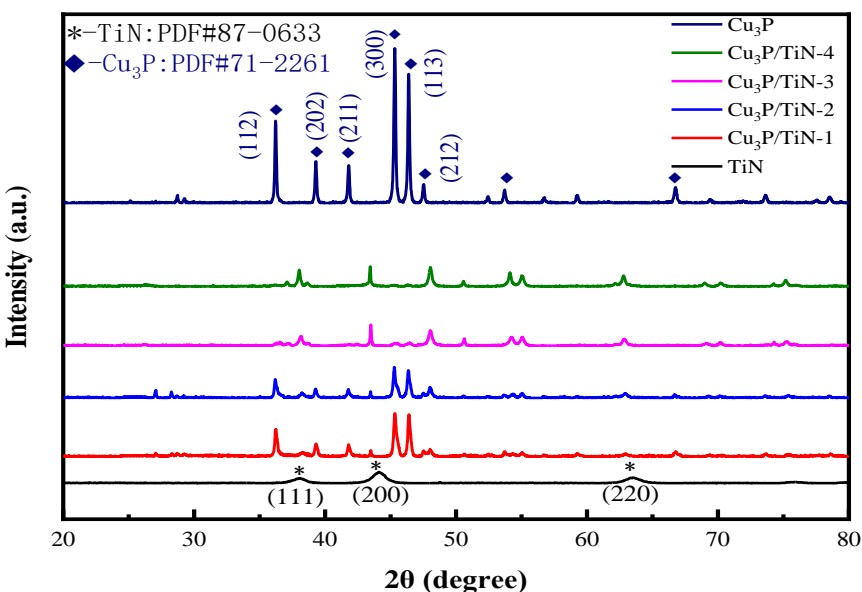

**Figure 1.** XRD patterns of the prepared photocatalysts.

2.1.2. SEM Images

The morphological structures of bare TiN, bare Cu₃P, and Cu₃P/TiN—x samples were detected by scanning electron microscopy (SEM). In Figure 2a, TiN shows a flocculent structure with a rough and fluffy surface, leading to a large specific surface area [30]. In Figure 2f, it can be seen that Cu₃P particles are uniform and exist in clusters. The SEM images of Cu₃P/TiN—1, Cu₃P/TiN—2, Cu₃P/TiN—3, and Cu₃P/TiN—4 were shown in Figure 2b–e sequentially. Cu₃P particles were densely loaded on the surface of TiN. After Cu₃P loading, the flocculent profile of TiN can still be identified clearly, indicating that the structure of TiN was quite stable [31]. TiN and Cu₃P were tightly bound together and provided more active sites, which will be further verified in the subsequent sections. These results indicate that Cu₃P was successfully loaded on the TiN surface, and the Cu₃P/TiN—x composites were successfully fabricated.

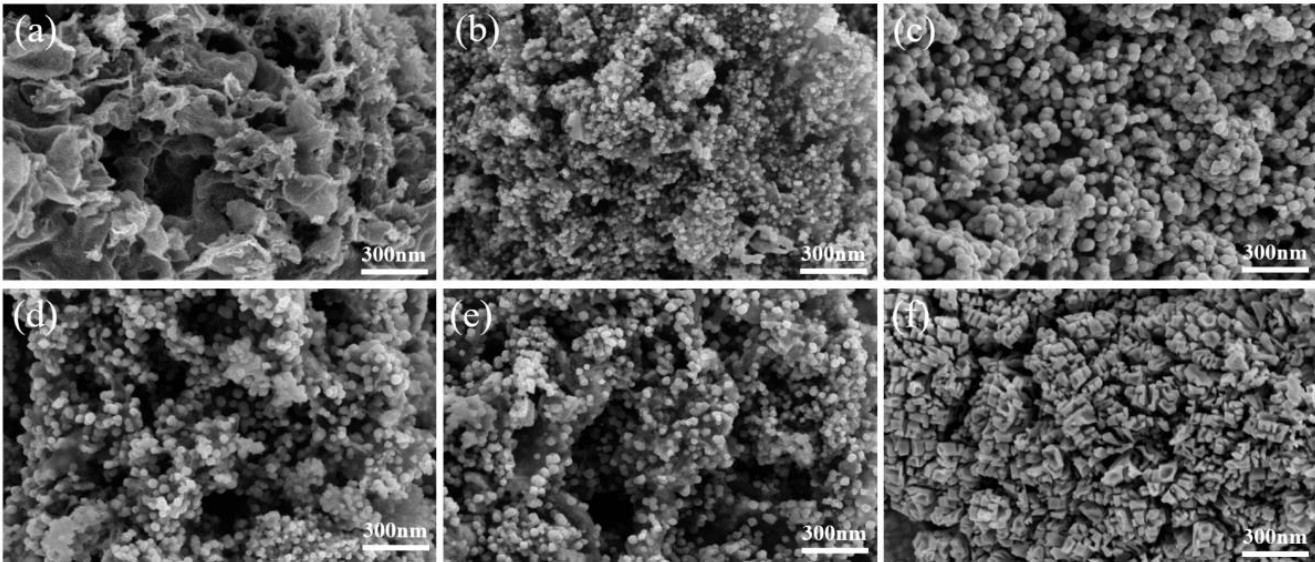

**Figure 2.** SEM images of (**a**) bare TiN, (**b**–**e**) Cu₃P/TiN—1~4, (**f**) bare Cu₃P.

### 2.1.3. $N_2$ Adsorption—Desorption Isotherms

The specific surface area and pore structure of the prepared photocatalysts were collected by $N_2$ adsorption and desorption isotherms. As shown in Figure 3a, the isotherms of $Cu_3P$, TiN, and $Cu_3P/TiN$—x composites were a classic type IV [32]. $Cu_3P$, TiN, and the $Cu_3P/TiN$—4 were typical H3 hysteresis loops [33]; however, the $Cu_3P/TiN$—1~3 composites were typical H4 hysteresis loops, indicating that $Cu_3P$, TiN, and the $Cu_3P/TiN$—1~4 composites all have mesoporous structures. The H3 and H4 hysteresis loop isotherms have no obvious saturation adsorption plateau; these data indicated that the pore structure of $Cu_3P/TiN$—x composites was very irregular, and $Cu_3P/TiN$—x composites were mixed microporous and mesoporous material. Besides, the corresponding pore size distributions of TiN, $Cu_3P$, and $Cu_3P/TiN$—x are also shown in Figure 3b. Table 1 has collected all of the data for the prepared materials, including BET surface area, pore volume, and average pore size. The BET surface area of $Cu_3P$ was calculated to be 0.8 $m^2\ g^{-1}$, and the pore size was 5.4 nm. BET surface area and pore size of TiN were 111 $m^2\ g^{-1}$ and 10.2 nm, and those of $Cu_3P/TiN$—1 were 55 $m^2\ g^{-1}$ and 5.0 nm, respectively. Noticeably, the BET—specific surface areas were 120, 123, and 142 $m^2\ g^{-1}$ for the $Cu_3P/TiN$—2, $Cu_3P/TiN$—3, and $Cu_3P/TiN$—4, respectively. As shown in Figure 3b, the pore size of $Cu_3P/TiN$—2, $Cu_3P/TiN$—3, and $Cu_3P/TiN$—4 was 4.5, 6.6 and 6.8 nm, respectively. In general, the high surface area and mesoporous structure of photocatalytic materials were beneficial in improving their adsorption capacity and photocatalytic activity. These will be further demonstrated in the next experiment.

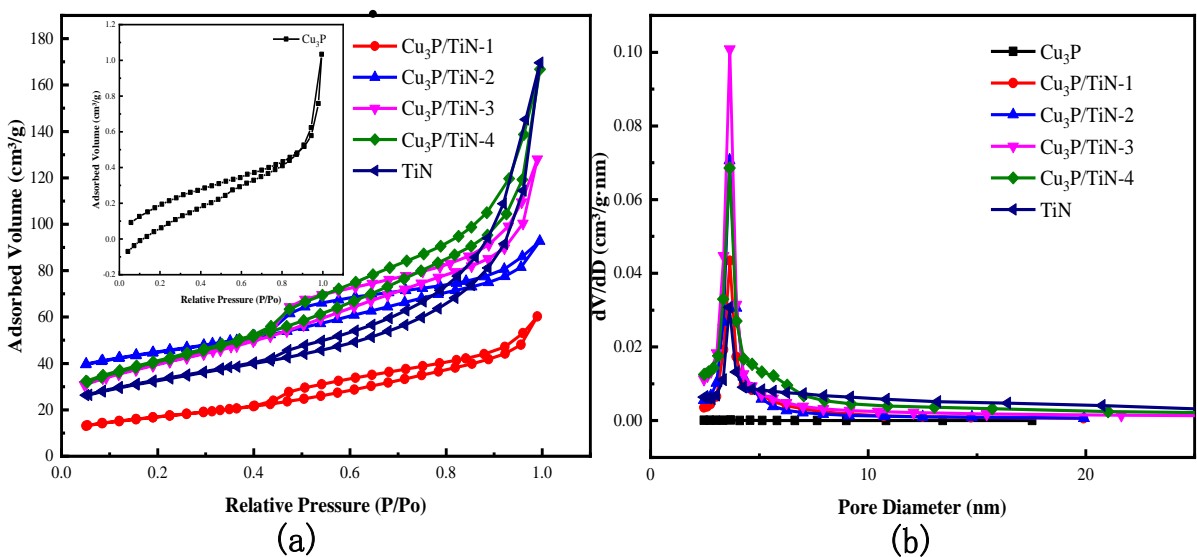

**Figure 3.** $N_2$ adsorption–desorption isotherms (**a**), BJH pore size distributions of $Cu_3P$, TiN, and $Cu_3P/TiN$—x samples (**b**).

**Table 1.** Specific surface area and pore size of different samples.

| Sample | Specific Surface Area ($m^2$/g) | Pore Volume ($cm^3$/g) | Average Pore Size (nm) |
|---|---|---|---|
| $Cu_3P$ | 0.8 | 0.001 | 5.4 |
| $Cu_3P/TiN$—1 | 55 | 0.09 | 5.0 |
| $Cu_3P/TiN$—2 | 120 | 0.16 | 4.5 |
| $Cu_3P/TiN$—3 | 136 | 0.20 | 6.6 |
| $Cu_3P/TiN$—4 | 142 | 0.26 | 6.8 |
| TiN | 111 | 0.26 | 10.2 |

## 2.2. Optical Property

### 2.2.1. Map of DRS

The optical properties of the samples were studied using a UV—vis diffuse reflectance spectrometer. The results are shown in Figure 4a. Bare $Cu_3P$ has an absorption ranging between 200 and 750 nm, while the prepared $Cu_3P/TiN$—x composites have a stronger absorption between 200 and 800 nm. Compared with bare $Cu_3P$, the absorption edge of $Cu_3P/TiN$—x composites shifted to a long—wave direction (red shift) after adding TiN [34]. In addition, with the increase in the TiN ratio, the light absorption was gradually enhanced, further proving that the combination of $Cu_3P$ and TiN was a good strategy [35]. The band—gap energy of $Cu_3P$ and $Cu_3P/TiN$—x can be obtained by the following equation:

$$(\alpha h v)^{(1/n)} = A(h v - E_g)$$

where $\alpha$, h, and v represent the absorption coefficient, Planck's constant, and optical frequency, respectively. A and $E_g$ represent constant and band—gap energy [36]. In general, the value of n is 1/2 for direct band—gap material, and the value of n is 2 for indirect band—gap material [37]. The value of $E_g$ can be determined by plotting the relationship between $(\alpha h v)^{1/2}$ and hv [38]. The Tauc plots (inset in Figure 4b) revealed that the band—gap energy of $Cu_3P$ was 1.61 eV, for $Cu_3P/TiN$—2, the band gap was 2.01 eV. Due to TiN doping, the band gap of $Cu_3P/TiN$—2 (2.01 eV) was wider than the intrinsic band gap of $Cu_3P$ (1.61 eV). The small band gap of $Cu_3P$, although highly beneficial in terms of its photo—excitability, experienced a high recombination rate. Electrons and holes, once generated, recombined at a very fast rate, rendering them unusable for mediating any photoredox reaction. TiN provides the surface for excited electrons of $Cu_3P$ and prevents their recombination; the wider band gap of the $Cu_3P/TiN$—2 composite has improved the redox ability.

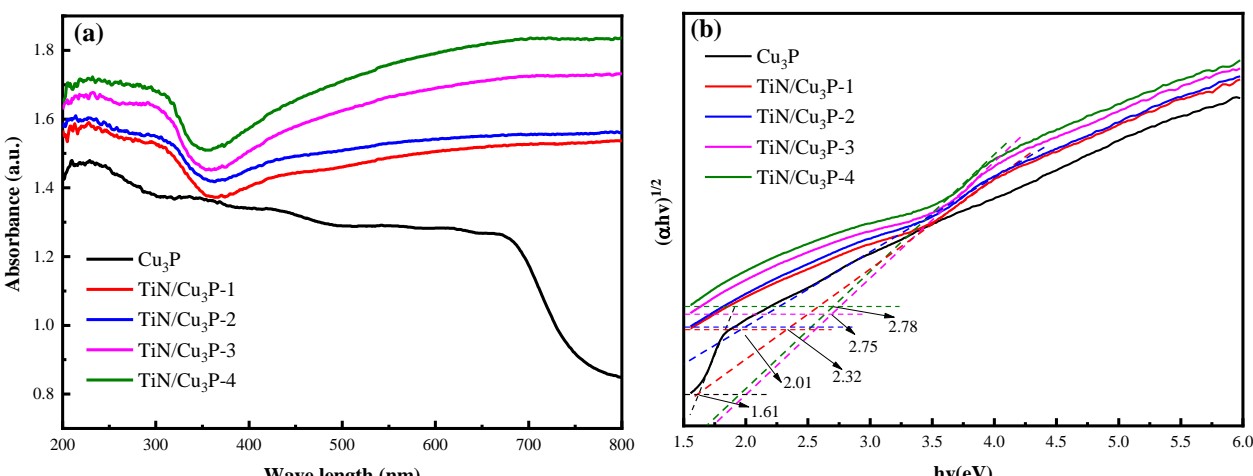

**Figure 4.** (**a**) UV—vis diffuse reflectance spectra of $Cu_3P$ and $Cu_3P/TiN$—x composites; (**b**) Tauc plots obtained for the indicated samples.

### 2.2.2. Electrochemical Impedance

The diameter of the Nyquist plot circle represents the electron transfer resistance ($R_{ct}$) [39–41]. The EIS patterns of $Cu_3P$ and $Cu_3P/TiN$—x composites have been given in Figure 5. The $Cu_3P$ owns the largest $R_{ct}$, about 60 Ω. Besides, the $R_{ct}$ of $Cu_3P/TiN$—1~4 composites were all reduced compared to $Cu_3P$. It was demonstrated that the introduction of TiN does improve the conductivity of the composites. Among them, $Cu_3P/TiN$—2 composite showed the smallest $R_{ct}$, only 30 Ω. The $R_{ct}$ of $Cu_3P/TiN$—1, $Cu_3P/TiN$—3, and $Cu_3P/TiN$—4 were 35, 40, and 45 Ω, respectively. The order of the electrochemical impedance values for $Cu_3P/TiN$—x composites was $Cu_3P/TiN$—2 < $Cu_3P/TiN$—1 < $Cu_3P/TiN$—3 < $Cu_3P/TiN$—4 < $Cu_3P$, indicating that $Cu_3P/TiN$—2 composite has the best

electron transfer ability and the highest ability to reduce photogenerated electron—hole pairs recombination. All these properties contributed to the enhanced photocatalytic activity of the catalyst and improved the removal efficiency of SMX.

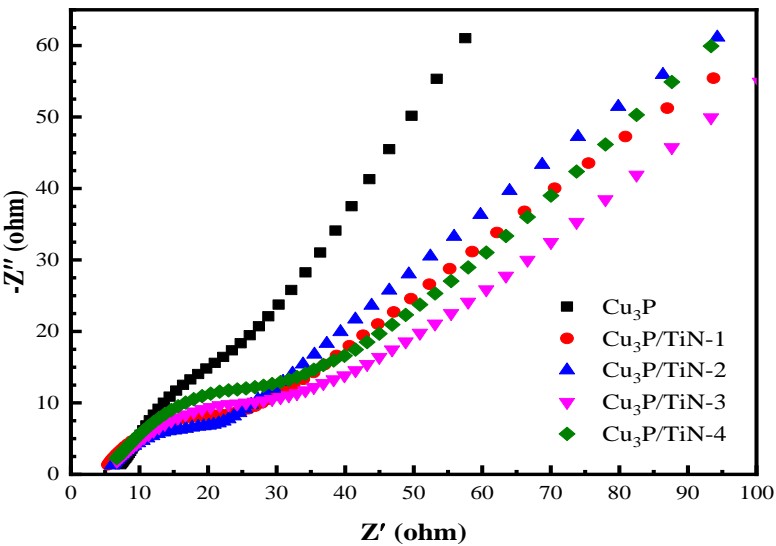

**Figure 5.** Nyquist plot of $Cu_3P$ and $Cu_3P/TiN$—1~4.

### 2.3. Performance of Photocatalysts

#### 2.3.1. Activity of the Photocatalyst

In order to investigate the photocatalytic activity of different photocatalysts, photocatalytic experiments were performed under visible light conditions. Figure 6a shows that all samples exhibit a certain removal rate for SMX under 30 min adsorption and 120 min photocatalysis. After 30 min dark reaction, all the systems reached adsorption equilibrium, and the adsorption efficiency of $Cu_3P$ and TiN for SMX reached 26% and 50%, respectively. After loading $Cu_3P$ particles on the surface of TiN, the adsorption ability of the $Cu_3P/TiN$—1~4 composites for SMX was appropriately strengthened, consistent with the trend presented by the BET results, i.e., $Cu_3P/TiN$—1 (55%) < $Cu_3P/TiN$—2 (63%) < $Cu_3P/TiN$—3 (66%) < $Cu_3P/TiN$—4 (68%). After the next 120 min illumination, $Cu_3P$ shows a nearly 26% removal rate for SMX, but TiN has almost no removal efficiency for SMX. For $Cu_3P/TiN$—1~4, the order of removal rates was: $Cu_3P/TiN$—4 (12%) < $Cu_3P/TiN$—3 (13%) < $Cu_3P/TiN$—2 (27%) < $Cu_3P/TiN$—1 (30%). Combined with the conclusion in Figure 6a, it can be revealed that the $Cu_3P/TiN$—2 composite finally showed the best removal rate (90%) of SMX owning to the synergistic action of adsorption and photocatalysis. The introduction of TiN not only increases the SMX adsorption of the $Cu_3P/TiN$—2 but also provides more active sites for the photocatalytic reaction, ultimately improving the removal efficiency of SMX on the whole. Here, it can be considered that an optimal dynamic process of adsorption—photocatalytic degradation—adsorption was realized for $Cu_3P/TiN$—2, eventually realizing a high removal rate for SMX.

Figure 6b shows the photodegradation data according to the quasi—first—order kinetic model ($-\ln (C/C_0) = kt$) [42,43], where k and t represent the reaction rate constant and irradiation time, $C_0$ represents the initial concentration of SMX, and C represents the concentration of SMX at time t [44]. Figure 6c shows the bar graph of the rate constants (k) of TiN, $Cu_3P$, and the $Cu_3P/TiN$—1~4 for SMX degradation. The rate constants (k) of $Cu_3P/TiN$—x were 0.009, 0.0102, 0.004, and 0.0046 $min^{-1}$, respectively, which were all larger than that of the bare $Cu_3P$ (0.0036 $min^{-1}$) and TiN (0.0008 $min^{-1}$). Thereinto, $Cu_3P/TiN$—2 owns the largest k, which was 2.83 times and 12.75 times as fast as $Cu_3P$ and TiN. The results showed that the $Cu_3P/TiN$—x composites could effectively improve the SMX removal efficiency, it was also demonstrated in Table 2. The $Cu_3P/TiN$—2 achieved the optimal degradation capacity. In the process of photocatalytic degradation of SMX

by $Cu_3P/TiN$—2, the absorption spectra of $Cu_3P/TiN$—2 for SMX at different times was shown in Figure 6d. With the gradual extension time of the visible light radiation, the characteristic absorption wavelength of SMX at 265 nm has changed significantly. It confirmed that the SMX gradually breaks down into small molecules as the extension time of visible light radiation. In other words, in addition to SMX, new substances have appeared in the solution system.

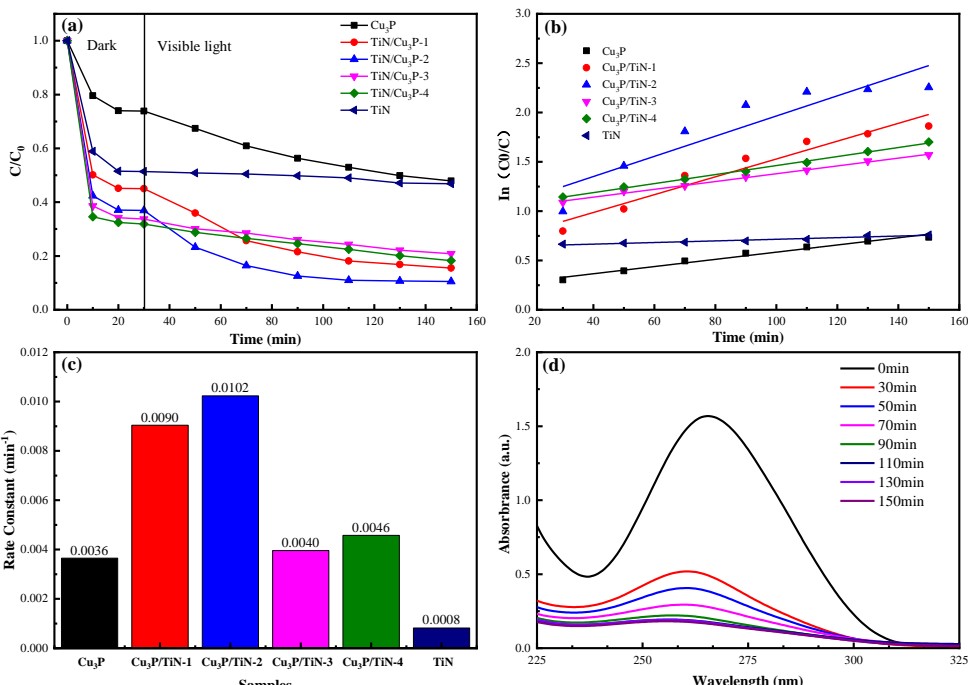

**Figure 6.** (**a**) Photocatalytic degradation curves of SMX over $Cu_3P$, TiN, and $Cu_3P/TiN$—1~4, (**b**) the pseudo—first—order kinetics curves, (**c**) rate constants, (**d**) SMX absorption spectra of $Cu_3P/TiN$—2 at different time points.

**Table 2.** Comparison of photodegradation effects of SMX under different catalysts.

| Catalyst | SMX Amount (mg/L) | Pollutant Concentration (g/L) | Rate Constant (min$^{-1}$) | Dark (min) | Illumination (min) | Removal Rate | Reference |
|---|---|---|---|---|---|---|---|
| g—N—$TiO_2$ | 1 | — | 0.0061 | 30 | 250 | 85% | [21] |
| $Ag_3PO_4/Bi_4Ti_3O_{12}$ | 5 | — | 0.0372 | 10 | 50 | 80% | [10] |
| $Cu_3P/BiVO_4$ | 0.5 | 0.75 | 0.0217 | 10 | 120 | 77% | [45] |
| $Bi_2O_3$ | 10 | 1 | 0.009 | 30 | 200 | 52% | [46] |
| p(HEA/NMMA)—CuS | 50 | 2 | — | 30 | 210 | 75% | [47] |
| $Cu_3P/TiN$ | 50 | 0.5 | 0.0102 | 30 | 120 | 90% | This Work |

### 2.3.2. Stability and Reusability

In order to verify the cyclic stability and reusability of the photocatalysts in practical application, recycling experiments were studied [48]. Therefore, the photocatalytic activity of $Cu_3P/TiN$—2 through five successive cycles was tested. After a 2-h illumination experiment, the photocatalyst was separated from the solution by filtration and washed with deionized water, and then used in the subsequent photocatalytic experiments for another four cycles. As shown in Figure 7a,b, for $Cu_3P/TiN$—2 catalyst, the removal efficiency only decreased by 7% after five cycles. The results showed that the $Cu_3P/TiN$—2 has good stability and reusability, which can effectively reduce the production cost of catalysts in practical applications.

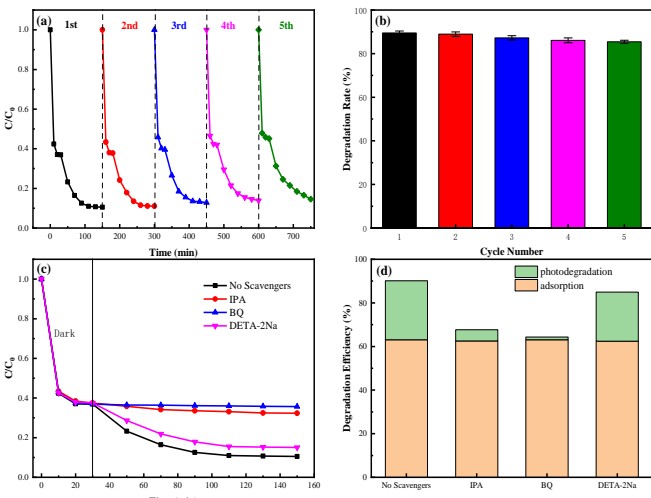

**Figure 7.** (**a**) The reusability of Cu$_3$P/TiN—2 photocatalyst for degradation of SMX, (**b**) degradation rate of SMX after recycling, (**c**) degradation of SMX by Cu$_3$P/TiN—2 with different capture agents, (**d**) histogram of degradation efficiency of different capture agents.

The photocatalytic reaction mechanism of SMX was studied by adding an active oxidant species capture agent at the photocatalytic reaction stage. It can be seen from Figure 7c,d, with the addition of IPA and BQ as capture reagents for ·OH and ·O$_2^-$, the removal efficiency of SMX decreased to 67% and 64%, respectively, much lower than the blank experiment (90%). This strong inhibition suggests that ·OH and ·O$_2^-$ play a major role in SMX degradation, of which the role of ·O$_2^-$ is more obvious. However, when EDTA—2Na was used as the capture agent during the degradation experiments, the removal efficiency of SMX decreased from 90% to 85%, indicating that h$^+$ played a less significant role in the photocatalytic degradation of SMX.

### 2.3.3. Photocatalytic Mechanism

The photocatalytic degradation mechanism of Cu$_3$P/TiN—x composites was depicted in the schematic diagram in Figure 8. Under visible light, the photogenerated electrons migrate into the conduction band of Cu$_3$P. Then, the photogenerated electrons inject into TiN and react with O$_2$ on the surface of TiN to form ·O$_2^-$. The holes left in the valence band of Cu$_3$P will react with H$_2$O to form ·OH. ·O$_2^-$ and ·OH are involved in the degradation of SMX. It can be considered that TiN promotes the separation of photogenerated electrons and holes, and the LSPR effect on the TiN surface enhances the visible light absorption capacity of the catalyst system, which is consistent with the conclusion in Figure 4.

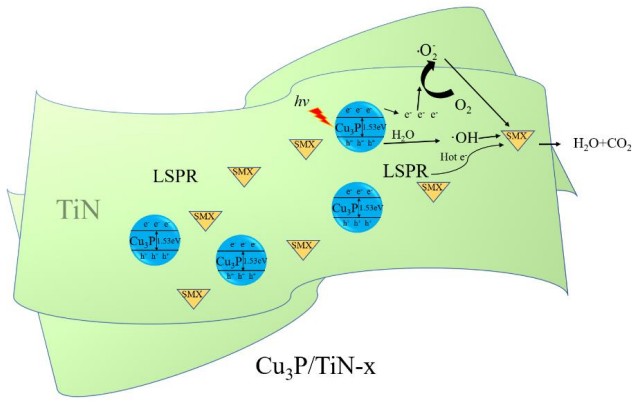

**Figure 8.** Photocatalytic mechanism under Cu$_3$P/TiN—x visible light irradiation.

## 3. Materials and Methods

### 3.1. Chemicals

All chemical reagents were analytical grade purity and were used without further treatment. Copper powder (Cu), red phosphorus ($P_4$), Ethylene diamine tetra acetic acid disodium salt (EDTA—2Na), benzoquinone (BQ), isopropyl alcohol (IPA) and sulfamethoxazole were purchased from Shanghai Aladdin Biochemical Technology Co., Ltd. (Shanghai, China).

### 3.2. Preparation of Photocatalysts

A synthesis of $Cu_3P$: 0.189 g copper powder and 0.3097 g red phosphorus was fully ground. Then, the mixture was placed in a beaker and 80 mL of deionized water added. After stirring continuously for 30 min, it was transferred to a 100-mL reaction kettle and then put into a 200 °C oven for continuous heating for 24 h. Afterward, the above—mentioned mixture was washed by carbon disulfide, ethanol, and deionized water in turn [12]. Finally, the mixture was dried in an oven at 60 °C for 8 h.

The synthesis of g—$C_3N_4$: 20 g urea was placed in an alumina crucible and calcined in a muffle furnace at 550 °C for 4 h at a heating rate of 2 °C $min^{-1}$ [47].

A synthesis of TiN: 1.20 g g—$C_3N_4$ plus 0.20 g hexadecyl trimethyl ammonium bromide (CTAB) was dissolved in 10 mL deionized water. This mixture was labeled as A and put into the ultrasound device at 50 °C and ultrasonic processing for 30 min. Next, 2.20 g of zinc acetate dihydrate and 2.65 mL titanium isopropyl oxygen were mixed and then dissolved in 3.33 mL of acetic acid ($CH_3COOH$) to form a mixture. This mixture was labeled B (Zn:Ti = 1:0.9). Finally, B was treated at 50 °C for 10 min by sonication. Then, solution A was mixed into B and the mixture under ultrasonic conditions for 10 min to form a uniform suspension. The above—mentioned mixture was placed in an oven at 50 °C and heated at a constant temperature for 3 h. Then, it was made into transparent sol [25]. In order to completely remove excess acetic acid, the sol was put in an oven and heated at 120 °C for 10 h at a constant temperature. Finally, the product was transferred to the crucible and calcined under the muffle furnace at 900 °C for 2 h.

Synthesis of $Cu_3P$/TiN—x: Firstly, 0.07624 g copper powder, 0.124 g red phosphorus, and a certain amount of TiN were fully ground. Secondly, the mixed powder was put into a beaker and 80 mL of deionized water was added. The uniform suspension was obtained after 30 min. Thirdly, the suspension was transferred into a 100-mL reaction kettle and treated in an oven at 200 °C for 24 h. Fourthly, the suspension was centrifuged and washed with deionized water and ethanol several times. Finally, in order to obtain $Cu_3P$/TiN—x composites, the uniform suspension was dried in an oven and heated at a constant temperature of 120 °C for 10 h. According to the different Cu/Ti molar ratios, i.e., 1/0.5, 1/1, 1/5, and 1/10, respectively, $Cu_3P$/TiN—x were expressed as $Cu_3P$/TiN—1, $Cu_3P$/TiN—2, $Cu_3P$/TiN—3, and $Cu_3P$/TiN—4.

### 3.3. Characterization Methods

To understand their physical characteristics, samples were characterized with the help of the existing functional groups, the absorption intensity of visible light, the efficiency of degradation pollutants, and the stability of each catalyst material. The phase composition of the prepared samples was analyzed by X—ray diffraction (XRD) (Brucker test in German) on a D/MAX—2500 diffractometer. To analyze the microscopic physical morphology characteristics of the samples prepared in the experiment, the scanning electron microscope (SEM) of Hitachi (S—4800) was used. X—ray photoelectron spectroscopy (XPS) was measured on an ESCALAB 250 X—ray photo spectrometer (Thermo Fischer, Waltham, MA, USA). The optical absorption characteristics of the prepared samples were investigated through an ultraviolet—visible diffuse reflectivity spectrometer (UV—vis DRS) of Shimadzu UV 2600 model (Hitachi, Japan). Moreover, the wavelength range of the device was maintained between 250 and 800 nm, and the ultraviolet—visible diffuse reflectivity spectrometer with barium sulfate was used as the reference. $N_2$ adsorption and desorption isotherms of the composites were tested by Quantachrome—NOVA 2200e—Surface Area

and Pore Size Analyzer, and the surface area and pore size characteristics of the composites were calculated using BET theory and BJH theory models, respectively.

Finally, the electrochemical impedance spectroscopy (EIS) test was carried out on the electrochemical workstation of Chenghua Instrument Co. Shanghai, China (CHI660E). The Pt plate and saturated Ag/AgCl electrode were used as auxiliary electrodes and reference electrodes; the prepared samples served as working electrodes, 0.5 M $H_2SO_4$ aqueous solution was used as the electrolyte. The electrode sample preparation process was as follows: 0.20 g polyvinylidene fluoride (PVDF) binder was mixed with a 0.02 g catalyst sample and 0.02 g carbon black in a small glass jar. Then we added 500 μL N—methyl pyrrolidone to the mixture and stirred. After a few hours, the paste liquid was consistently coated on a copper sheet $10 \times 20$ mm in size, and the other side of the copper sheet was covered with insulating glue. The open—circuit potential of the electrode was then tested, and the initial voltage was set according to the open—circuit potential. The open—circuit potential of $Cu_3P$ tested in this experiment was 0.008 V, and the open—circuit potentials of $Cu_3P/TiN$—1, 2, 3, and 4 samples were 0.010 V, 0.025 V, 0.043 V, and 0.057 V, respectively. The initial voltage E was set according to the open—circuit potentials of the tested samples, and the High frequency = $10^{+5}$ Hz, Low frequency = 1 Hz.

### 3.4. Photocatalytic Activity

To evaluate the activity of different catalysts, experiments on photocatalytic degradation were carried out with SMX as a substrate. All photocatalytic degradation experiments were carried out under the condition of a 1000 w xenon lamp [49], and the reaction temperature of the system was controlled at 25 °C by the circulating cooling device. To ensure the uniform dissolution and dispersion of SMX, the SMX solution (50 mg $L^{-1}$) was ultrasonically treated for 20 min, and 20 mg photocatalyst ($Cu_3P$, TiN, or $Cu_3P/TiN$—x) was dispersed into 40 mL SMX solution (50 mg $L^{-1}$) to carry out the next photocatalytic degradation experiment. The catalyst concentration was 0.5 g $L^{-1}$. Before light experiments, the mixed liquid was stirred under dark conditions for 30 min to reach the equilibrium of adsorption and desorption, excluding the strong adsorption effect of the catalyst on SMX during the photocatalytic experiment. Then researchers turned on the light source of the wavelength needed for the experiment and conducted the photocatalytic degradation experiment, in which 3 mL samples were taken every 20 min and diluted with 3 mL deionized water. In order to remove the precipitate, the diluted samples were centrifuged twice at 10,000 r $min^{-1}$. Then, the supernatant of the sample was removed, and the concentration of SMX among the catalytic materials was measured at 265 nm by a UV—visible spectrophotometer. The whole experimental on photocatalytic degradation of SMX lasted for 120 min.

### 4. Conclusions

In short, the novel $Cu_3P/TiN$ photocatalyst was successfully synthesized by direct loading $Cu_3P$ on the surface of TiN through thermal annealing. The removal rate of the $Cu_3P/TiN$—2 composite for SMX was significantly enhanced and was up to 90%. The improved removal efficiency of SMX attributes to the efficient adsorption—photocatalytic degradation dynamic equilibrium mechanism. The improved removal efficiency is supported by the significantly reduced recombination of photo—excited electron—hole pairs and supplied more active sites. In addition, $Cu_3P/TiN$—2 exhibited excellent cyclic stability and reusability, and the removal efficiency only decreased by 7% after five cycles. This work provides a new idea and strategy for the efficient removal of organic pollutant SMX.

**Author Contributions:** H.S.: Funding acquisition, Writing—review & editing. X.Y.: Investigation, Experiments, and Writing—original draft, proofreading, and editing. S.L.: Writing—review & editing. Y.Z.: Conceptualization, Methodology, Investigation, Writing—review & editing. T.Z.: Investigation, Data curation, and plotting Formal analysis. L.J.: Funding and reviewing. All authors have read and agreed to the published version of the manuscript.

**Funding:** This study was funded by the National Natural Science Foundation of China (NSFC 21606150), the Bureau of Huzhou Municipal Science and Technology (2021ZD2043, 2021ZD2003), and China Denmark International Cooperation Project (2022YFE0115800).

**Data Availability Statement:** No new data were created or analyzed in this study. Data sharing is not applicable to this article.

**Conflicts of Interest:** The authors declare no conflict of interest.

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
