# Peer review of "Photocatalytically Active Semiconductor Cu3P Unites with Flocculent TiN for Efficient Removal of Sulfamethoxazole"

_catalysts, doi:10.3390/catal13020291_

Round 1

Reviewer 1 Report

This paper describes the adsorption and photodegradation of sulfamethoxazole using Cu3P-TiN composite. The composite exhibited higher activity than Cu3P or TiN alone. However, in order to claim the usefulness of this composite, it should be compared to other typical photocatalysts, otherwise, it is not clear why this Cu3P/TiN composite was used. In addition, the statement “TiN can generate a plasma, and then the plasma fundamental oscillations can excite hot electrons…(L239)” is not clear to me. I couldn’t find any experimental results related to the statement.

Furthermore, although the authors claimed that “SEM images also demonstrated that TiN and Cu3P were attached together, and Cu3P/TiN-x composite was successfully prepared as well (L329)”, this claim cannot be accepted based on the SEM images alone, and EDX analysis should at least be performed at higher magnification. Alternatively, TEM analysis is required.

Overall, this manuscript is not suitable for publication due to the lack of appropriate experiments, analysis, and discussion. Additional comments are below.

1) Check English usage.

2) Should know significant digits (or significant figures)

3) Fig.6(b) Incorrect fitting

4) L251. What is phosphating copper? L49. “Transition metal phosphates”? Phosphate and phosphide are different.

Reviewer 2 Report

In this manuscript, a heterostructure, Cu3P/TiN-x with visible optical response, was successfully prepared by simply calcination. The Cu3P/TiN-x composites remarkably removed the sulfamethoxazole in solution compared with Cu3P and TiN alone. Among the composites, the Cu3P/TiN-2 with 1:1 molar ratio of Cu: Ti reached 90% under dark adsorption for 30 min and subsequent photodegradation for 120 min. The enhanced performance of the Cu3P/TiN-x composites attributed to the introduced flocculent TiN with large specific surface area and high conductivity that provide more active sites and high electron transfer ability. Meanwhile, the strong corrosion resistance and chemical stability were also beneficial to the improved performance. The reaction mechanism proposed for capture experiments. Hence, I recommend the publication of this manuscript in Catalysts after minor revision.

1. In photocatalytic mechanism schematic diagram, the TiN are believed to be semiconductor to generate the electrons under light irradiation. Please confirm the VB and CB positions of both Cu3P and TiN to validate Cu3P/TiN-x composite is a heterostructure.

2. In Figure 4(b), the method of verify band gap is wrong. This article is for your reference to revise it. (10.1016/j.apcatb.2018.10.027)

3. Please add a photocurrent to further confirmed that the higher separation ratio of electron-hole pairs of Cu3P/TiN-x composite.

4. Some relative works should be cited. (10.1002/adfm.202100553; 10.1021/acsami.0c20919; 10.1039/D0TA00488J)

Round 2

Reviewer 1 Report

Unfortunately, the authors do not understand the meaning of significant digits. Considering significant digits is fundamental to science. Any experiments have experimental errors.

For example, in Table 1, the specific surface area for Cu3P/TIN-4 is described as 141.5937 m2 g-1. However, this value is impossible to be obtained if the same measurement is conducted again. 

Thus, the value should be written as 142 m2 g-1 when the significant digits (or significant figures) are considered. 

To summarize, the specific surface area, pore volume, and average pore size of Cu3P/TiN-4  should be  142 m2 g-1, 0.26 cm3 g-1, and 6.8 nm, respectively.

This should be also adapted to other materials.
